# Structural Characterization and Anti-Osteoporosis Effects of a Novel Sialoglycopeptide from Tuna Eggs

**DOI:** 10.3390/md21110573

**Published:** 2023-10-31

**Authors:** Shiwei Hu, Xiaofeng Wan, Hongli Zhu, Huicheng Yang

**Affiliations:** 1National Engineering Research Center for Maine Aquaculture, Zhejiang Ocean University, Zhoushan 316022, China; hushiweihai@163.com (S.H.); wanxiaofeng1998@163.com (X.W.); zhuhongli2020@163.com (H.Z.); 2Zhejiang Marine Development Research Institute, Zhoushan 316021, China

**Keywords:** sialoglycopeptide, tuna eggs, chemical structure, anti-osteoporosis

## Abstract

Several sialoglycopeptides were isolated from several fish eggs and exerted anti-osteoporosis effects. However, few papers have explored sialoglycopeptide from tuna eggs (T-ES). Here, a novel T-ES was prepared through extraction with KCl solution and subsequent enzymolysis. Pure T-ES was obtained through DEAE-Sepharose ion exchange chromatography and sephacryl S-300 gel filtration chromatography. The T-ES was composed of 14.07% protein, 73.54% hexose, and 8.28% Neu5Ac, with a molecular weight of 9481 Da. The backbone carbohydrate in the T-ES was →4)-β-D-GlcN-(1→3)-α-D-GalN-(1→3)-β-D-Glc-(1→2)-α-D-Gal-(1→2)-α-D-Gal-(1→3)-α-D-Man-(1→, with two branches of β-D-GlcN-(1→ and α-D-GalN-(1→ linking at o-4 in →2,4)-α-D-Gal-(1→. Neu5Ac in the T-ES was linked to the branch of α-D-GlcN-(1→. A peptide chain, Ala-Asp-Asn-Lys-Ser*-Met-Ile that was connected to the carbohydrate chain through O-glycosylation at the –OH of serine. Furthermore, in vitro data revealed that T-ES could remarkably enhance bone density, bone biomechanical properties, and bone microstructure in SAMP mice. The T-ES elevated serum osteogenesis-related markers and reduced bone resorption-related markers in serum and urine. The present study’s results demonstrated that T-ES, a novel sialoglycopeptide, showed significant anti-osteoporosis effects, which will accelerate the utilization of T-ES as an alternative marine drug or functional food for anti-osteoporosis.

## 1. Introduction

Osteoporosis is a systemic bone disease characterized by bone microstructure destruction [1]. It is caused by a decrease in bone density and a deterioration of bone microstructure, leading to bone mass reduction, bone strength decrease, and bone brittleness increase [2,3]. Resulting from the high global prevalence and disability, the World Health Organization defines osteoporosis as the one of the three major senile diseases and frequently occurring disease in the world [4]. It was reported that there were 158 million people aged over 50 who suffered from osteoporosis worldwide in 2010 and this number would double by 2040 [5]. In China, it was estimated that the age-standardized prevalence of osteoporosis was 6.46% and 29.13%, respectively, for men and women aged 50 years and older [6], and the number of osteoporosis or bone loss patients would account for 533 million people in 2050 [7]. Therefore, effective treatment of osteoporosis has become one of the important and significant research areas.

Sialoglycoprotein/sialoglycopeptide is composed of sialic acids (5-*N*-acetyl-neuraminic acid (Neu5Ac) or 5-*N*-glycolyl-neuraminic acid (Neu5Gc)), carbohydrate chains, and protein/peptide chains. In the compound, sialic acids are located at the terminal monosaccharide, and the hydroxyl of anomeric carbon in carbohydrates is linked to the hydroxyl (O-glycopeptide bond) or the acylamino (*N*-glycopeptide bond) of specific amino acids in protein/peptides [8]. Some papers have reported that several sialoglycoprotein/sialoglycopeptide were isolated and characterized from fish eggs, including rainbow trout, medaka fish, pacific herring, and *Gadus morhua* [9,10,11,12,13]. Moreover, the structure of sialoglycoprotein/sialoglycopeptide was distinguished between the various fish species, showing the specific carbohydrate chain, peptide chain, glycopeptides bone, and the kinds of sialic acids. Several studies also proved that sialoglycoprotein/sialoglycopeptide from fish eggs exhibited anti-osteoporosis activities [12,14].

As one of the world’s three major nutritious fishes, tuna is a deep-sea economic fish with annual catches exceeding 6 million tons. Tuna eggs contain abundant protein, phospholipids, unsaturated fatty acids, etc. Our previous experiments proved that the concentration of sialoglycopeptide from tuna eggs (T-ES) accounts for 4% of the egg. Two antioxidative peptides were isolated from tuna eggs [15]. However, there are few reports on T-ES and its bioactivities. In this study, a novel sialoglycopeptide was isolated and purified from tuna eggs, and its structure profile was clarified. Furthermore, the anti-osteoporosis effects of T-ES were also investigated in senescence-accelerated mouse prone (SAMP6) mice. This research will provide a theoretical basis for application in the prevention and treatment of senile osteoporosis.

## 2. Results and Discussion

### 2.1. Isolation and Purification of T-ES and General Property Description

As shown in Figure 1A, four fractions were obtained through polarity separation with NaCl. According to the phenol–sulfuric acid method, T-ES existed in fraction 1 (W1, 9.36%). Purified T-ES was separated using a sephacryl S-300 column and its yield was 13.52%. The weight-average molecular weight of T-ES was 9481 Da (Figure 1B).

### 2.2. Chemical Composition of T-ES

Figure 1C shows that Neu5Ac existed in T-ES, accounting for 8.28% of its weight. There was 73.54% of hexose found in the T-ES using the phenol–sulfuric acid method, and 14.07% of protein content found using the automatic kjeldahl nitrogen analyzer. It was reported that sialoglycopeptide from *Gadus morhua* eggs was composed of 16.2% protein, 63.7% hexose, and 18.6% Neu5Ac [16]. As shown in Figure 1D and Table 1, the monosaccharides of the T-ES comprised Fuc:GalN:GlcN:Gal:Glc:Xyl:Man:GlcA (0.021:0.187:0.262:0.276:0.101:0.026:0.109:0.017), which was significantly different form the carbohydrates in the sialoglycopeptide in the *Gadus morhua* eggs (Man, GlcN and Gal) [16].

### 2.3. Glycosidic Bond Type of T-ES

*N*-glycosylation and O-glycosylation are the main linkages between carbohydrate and peptide. As shown in Figure 1E, T-ES with NaOH showed a peak at 20.44 min using the HPLC method, while T-ES without NaOH represented a peak at 17.71 min. These data suggest that T-ES was deduced to be a glycoprotein with O-glycosidic linkages between hydroxyamino acids and polysaccharides. The same peaks occurred at 24.75 min for the T-ES with and without PNGase F treatment (Figure 1F), suggesting an unchanged glycoprotein with *N*-glycosylation in T-ES suffered with PNGase F. These data indicate that T-ES has O-glycosylation between the peptides and carbohydrates, rather than *N*-glycosylation. It was reported that *N*-linked glycoproteins existed both in sialoglycopeptide from Gadus morhua eggs and in zebrafish egg chorion [16,17], while O-linked glycan units were present in neuraminic acid-rich glycoprotein from rainbow trout eggs [18]. These indicate that the glycosidic bond between carbohydrate chain and peptide chain is different among different species.

### 2.4. FT-IR, Methylation and NMR Spectroscopy Analysis for Carbohydrate Chain

We used FT-IR to examine the chemical functional groups of the carbohydrate chain in T-ES, and information about the absorption peaks from 3600 to 3200 cm^−1^ (characteristic peaks of carbohydrates) is displayed in Figure 2A. The stretching vibration of –OH had a prominent peak at 3392 cm^−1^, which was the characteristic peak of the carbohydrate [19]. The peak at 2935 cm^−1^ might be attributed to C-H stretching vibration [20]. The C=O stretching vibrations were responsible for the absorption peaks at 1733 cm^−1^ and 1540 cm^−1^ [19]. The variable vibration of C-H had a peak at 1457 cm^−1^ [21]. The C-O molecules were responsible for the prominent peaks at 1419 cm^–1^ and 1147 cm^–1^ [22]. The variable vibration of O-H had a peak at 1114 cm^–1^ [21]. The prominent peak at 873 cm^−1^ might be attributed to C-H variable vibration in the pyranose ring [20].

Methylation analysis data for the carbohydrate chain in T-ES, obtained using GC-MS, are shown in Figure 2B, Table 2 and Appendix A. At 11.817 min and 15.486 min, the methylated sugars of 2,3,4-Me_3_-Fuc and 2,3-Me_2_-Xyl were detected, demonstrating that T-ES contained Fuc-(1→ and →4)-Xyl-(1→. It was found that Glc-(1→, →3)-Glc-(1→, →4)-Glc-(1→, →6-Glc-(1→, →3,4)-Glc-(1→, →4,6)-Glc-(1→, and →3,6)-Glc-(1→ were present as fragment ions of 2,3,4,6-Me_4_-Glc, 2,4,6-Me_3_-Glc, 2,3,6-Me_3_-Glc, 2,3,4-Me_3_-Glc, 2,6-Me_2_-Glc, 2,3-Me_2_-Glc, and 2,4-Me_2_-Glc, respectively. At 17.925 min, 20.805 min, and 25.349 min, the characteristic fragment ions of 2,3,4,6-Me_4_-Man, 3,4,6-Me_3_-Man, and 2,3-Me_2_-Man suggested Man-(1→, →2)-Man-(1→, and →4,6)-Man-(1→ in T-ES, respectively. In addition, at 19.221 min, 32.637 min, and 34.700 min, the markers of GlcN-(1→, →3)-GlcN-(1→, and →3,6)-GlcN-(1→ were observed in the carbohydrate chain of the T-ES. The methylated sugar of 4,6-Me2-D-Gal-2-N and 4-Me1-D-Gal-2-N suggested →3)-GalN-(1→ and →3,6)-GalN-(1→ in T-ES.

The NMR method was used to analyze the chemical structure of the carbohydrate chain in the T-ES. In the HSQC spectra, the anomeric carbon signal was δ104.15, and the corresponding anomeric hydrogen signal was δ4.66 (Figure 2D). In the HH-COSY spectra (Figure 2C), the signals of H1-2, H2-3 and H3-4 were δ4.66/4.02, δ4.02/3.73, and δ3.73/3.70, respectively. We speculated that the signals of H1-H4 were δ4.06, δ4.02, δ3.73, and δ3.70, and the corresponding signals of C1-C4 were δ104.15, δ52.15, δ69.13, and δ83.08, respectively. Moreover, the signals of H5-6 were δ3.73/3.70, and the corresponding signals of C5-6 were δ70.71/62.28. These signals were assigned to →4)-β-D-GlcN-(1→. The anomeric carbon/hydrogen signals were δ94.88/5.00 (HSQC spectras). The signals of H1, H2, H3, H5, and H6 were δ5.00, δ4.14, δ3.75, δ3.73, and δ3.70, while the corresponding signals of C1, C2, C3, C5, and C6 were δ94.88, δ50.80, δ79.59, δ70.71, and δ62.28, respectively. These indicated that the signals could be attributed to →3)-α-D-GalN-(1→. Furthermore, the signals for anomeric carbon/hydrogen at δ102.98 and δ4.47 suggested that the glycosidic bond was →2,4)-α-D-Gal-(1→. The signals for anomeric carbon/hydrogen at δ104.50 and δ4.40 indicated that →3)-β-Glc-(1→ existed in the T-ES. A glycosidic bond was speculated as →3)-α-D-Man-(1→ from the signals for anomeric carbon/hydrogen at δ96.87 and δ5.05.

In the HMBC spectra (Figure 2E), the H1 in the glycosidic bond of →4)-β-D-GlcN-(1→ and the C3 in →3)-α-D-GalN-(1→ had correlated signals, suggesting that there was →4)-β-D-GlcN-(1→3)-α-D-GalN-(1→. present The H1 in →3)-β-Glc-(1→ and the C3 in →2,4)-α-D-Gal-(1→ had correlated signals, indicating that the T-ES presented →3)-β-Glc-(1→2,4)-α-D-Gal-(1→. The coherent signals of C1 in →2,4)-α-D-Gal-(1→ and of H2 in its internal glycosidic bond showed →2,4)-α-D-Gal-(1→2,4)-α-D-Gal-(1→. The correlated signals of C1 in →2,4)-α-D-Gal-(1→ and H3 in →3)-α-D-Man-(1→ manifested that there was glycosidic bond →2,4)-α-D-Gal-(1→3)-α-D-Man-(1→. In the NOESY spectra (Figure 2F), the H1 in →3)-α-D-GalN-(1→ and H3 in →3)-β-Glc-(1→ had coherent signals, suggesting →3)-α-D-GalN-(1→3)-β-Glc-(1→ was present in the T-ES. These data indicated that the backbone of the pylysaccharide chain in the T-ES was →4)-β-D-GlcN-(1→3)-α-D-GalN-(1→3)-β-D-Glc-(1→2)-α-D-Gal-(1→2)-α-D-Gal-(1→3)-α-D-Man-(1→.

In the HMBC spectra, the H1 in β-D-GlcN-(1→ and the C4 in →2,4)-α-D-Gal-(1→ had coherent signals, manifesting that there was β-D-GlcN-(1→2,4)-α-D-Gal-(1→. In the NOESY spectra, the coherent signals of H1 in α-D-GalN-(1→ and H4 in →2,4)-α-D-Gal-(1→ showed that the T-ES contained α-D-GalN-(1→2,4)-α-D-Gal-(1→. These data indicated that the branches of β-D-GlcN-(1→ and α-D-GalN-(1→ were linked to the backbone through o-4 in →2,4)-α-D-Gal-(1→. In addition, sialic acid (Neu5Ac and Neu5Gc) was attached at the end of Gal or GlcN [8], suggesting that the Neu5Ac in the T-ES was linked to the branch of β-D-GlcN-(1→.

### 2.5. MS Spectrometry for Peptide Chain

Amino acid sequences were analyzed using LC-MS/MS and the results are shown at Figure 3, Appendix A. There were five peptides in the T-ES. The area of peptide No. 4 greatly exceeded that of the other peptides. The sequence of the peptide No.4 chain in the T-ES was Ala-Asp-Asn-Lys-Ser*-Met-Ile. The experiments using NaOH treatment (Figure 1E) suggested that serine might exist in the O-glycosylation of hydroxyamino acids. This result was similar to that of the sialoglycoprotein isolated from rainbow trout eggs [10].

In summary, combining Neu5Ac, peptide, and glycan moieties, we speculated the structural outline of T-ES is as shown in Figure 4.

### 2.6. T-ES Increased Bone Density

Bone mineral density is the golden standard in the early diagnosis, therapeutic, and prognosis of osteoporosis [23]. As shown in Table 3, a high dosage of T-ES caused significant increases in the bone density of both the femur and of tibia by 40.75%, and 1.14-fold compared with SAMP mice (*p* < 0.05, Table 1). Moreover, tibial bone density in the T-ES-L group was markedly increased by 54.90% compared with the model group (*p* < 0.05). In addition, there were no significant differences of femoral bone density or tibial bone density between the alendronate (ALN) group and T-ES-H group (*p* > 0.05), suggesting that the effect of the high dosage of T-ES on enhancing bone density was equivalent to that of the ALN group. These indicate that T-ES can improve senile osteoporosis.

### 2.7. T-ES Enhanced Bone Biomechanical Properties

Bone biomechanical properties are of great significance in evaluating osteoporosis [24]. The max. load and max. deflection of both the femur and of tibia were significantly elevated with high dosage of T-ES compared to the SAMP6 mice (*p* < 0.05, Table 1). Moreover, a low dosage of T-ES caused remarkable increases in femoral max. load, tibial max. load, and tibial max. deflection (*p* < 0.05). These results indicate that T-ES can enhance bone strength and quality, and lessen fracture risk.

### 2.8. T-ES Improved Bone Microstructure

As shown in Table 3 and Figure 5, the SAMP6 mice treated with the high dosage of T-ES showed significant increases in femoral trabecular thickness, trabecular number, connectivity density, cortical bone thickness, and remarkable reductions in femoral trabecular separation and structural model index compared with the SAMP6 mice (*p* < 0.05). Similarly, the aforementioned parameters of the tibia were also improved in the T-ES-H group compared with the model group (*p* < 0.05). Furthermore, the low dosage of T-ES also dramatically increased femoral trabecular thickness and tibial cortical bone thickness, and decreased tibial trabecular separation and the structural model index (*p* < 0.05). Interestingly, the increase in femoral trabecular number was markedly greater in the T-ES-H group than that found in the ALN group (*p* < 0.05). These results indicate that T-ES can improve osteoporosis-induced destruction of bone microstructure, and promote the repair of bone tissue.

### 2.9. T-ES Elevated Serum Osteogenesis-Related Parameters

As an osteogenesis biomarker, BALP, an osteoblasts-secreted extracellular enzyme, can maintain bone growth and regeneration [25]. The high dosage of T-ES caused significant increase in serum BALP concentration in SAMP6 mice (*p* < 0.05). PICP is the essential protein for collagen synthesis in bone tissue, which is synthesized by osteoblasts [26]. PICP can reflect the activity of osteoblasts and the synthesis rate of type I collagen [27]. Both low and high dosages of T-ES increased serum PICP concentration in SAMP6 mice (*p* < 0.05). BGP can maintain bone mineralization rate, which has important value for diagnosing osteoporosis [28]. Serum BGP levels were dramatically elevated in both the T-ES-L group and T-ES-H group compared with the model group (*p* < 0.05). Interestingly, the effects of the high dosage of T-ES on increasing serum BGP were better than those of ALN. These results indicate that T-ES can elevate serum osteogenesis-related parameters.

### 2.10. T-ES Reduced Bone Resorption-Related Parameters in Serum and Urine

RANKL and OPG are pivotal factors stimulating osteoclasts differentiation and activization. The ratio of OPG to RANKL is closely related to the generation of osteoclasts, which resolves the rate of bone resorption [29]. In the present study, the ratio of OPG to RANKL is significantly decreased in T-ES-H group compared with the model group (*p* < 0.05). Cath-k, MMP-9, and CTX-I are the critical enzymes that effect bone resorption, which are secreted by mature osteoclasts [30]. The mice treated with the high dosage of T-ES showed remarkable serum Cath-k, MMP-9, and CTX-I concentrations compared with the SAMP6 mice. Xia et al. reported that sialoglycoproteins isolated from the eggs of *Carassius auratus* exhibited anti-osteoporotic activity through increasing the OPG/RANKL ratio [12]. Zhao et al. reported that *Gadus morhua* eggs’ sialoglycoprotein prevents high bone turnover by controlling the OPG/RANKL/TRAF6 pathway [14]. 

Ca and P are the dominant components of bone mineral [31], DPD is an important sector of collagen I [32]. The three parameters are the important indicators in evaluating bone loss [32]. In our study, urinary Ca, P, and DPD concentrations in the T-ES-H group were distinctly decreased compared with the model group. These results indicate that T-ES can mitigate osteoporosis through reducing bone resorption-related parameters.

## 3. Materials and Methods 

### 3.1. Extraction and Purification of the T-ES

Mature striped tuna eggs were gained from the Zhoushan International Aquatic Products Market (Zhoushan, China), and were captured from the South Pacific Ocean. The lipids were removed from the homogenized fish eggs using ethyl alcohol (1:10, m/V, 25 °C, 8 h), they were then lyophilized, and subsequently extracted in 0.5 M KCl to obtain crude sialoglycoprotein (4 °C, 2 h). The sialoglycoprotein was hydrolyzed using 1.5% alkaline protease in a water bath oscillator at 50 °C for 2 h, and then terminated enzymolysis at 100 °C for 10 min. After centrifugation at 7500× *g* for 15 min, the liquid supernatant was concentrated and lyophilized to collect crude T-ES. Pure T-ES was obtained using a DEAE-Sepharose fast flow ion and, subsequently, a sephacryl S-300 column with a differential refraction detector. T-ES was detected using the phenol–sulfuric acid method [33].

### 3.2. Determination of General Properties

Sialic acid in T-ES was analyzed by HPLC (LC-10A, Shimadzu, Tokyo, Japan) according to the method in [34]. Polysaccharide content was detected through the phenol–sulfuric acid method [33], and the monosaccharide composition of T-ES was analyzed using the ion chromatograph (ICS5000) method using a DionexCarbopacTMPA20 (3×150) column. Polypeptide content was determined using an automatic kjeldahl nitrogen analyzer (OLB9870, Olabo, Shenzhen, China) according to the method in [35]. The molecular weight of T-ES was determined using HPLC, according to the method in [36], with a BRT105-104-102 gel column (8 × 300 mm), a 0.05 M NaCl mobile phase (0.6 mL/min), 40 °C column temperature, 20 µL injection volume, and an RI-10A differential detector.

Glycosylation type of T-ES was analyzed using PNGase F enzymolysis or NaOH hydrolyzation. Briefly, T-ES was hydrolyzed with PNGase F enzyme (1%, pH 7.5, 40 °C, 5 h), and then detected using HPLC (as aforementioned HPLC methods) to verify whether *N*-carbohydrate–peptide linkages reside in T-ES or not. On the other hand, O-carbohydrate–peptide linkage was confirmed through NaOH treatment (5 M, 25 °C, 2 h) and subsequent HPLC detection (as aforementioned HPLC methods). The samples without PNGase F and NaOH treatment were used as the control.

### 3.3. FT-IR Spectroscopy Analysis

The O-carbohydrate chain of the T-ES was prepared with an NaOH treatment and collected with an Sep-Pak C18 chromatographic column. O-carbohydrate chain component (2 mg) was mixed KBr (200 mg), crushed, compressed into a pellet, and subsequently scanned using a Fourier transform–infrared spectrometer (FT-IR650, Gangdong, Tianjin, China), in the 4000–500 cm^−1^ wavenumber region, at resolution of 4 cm^−1^, for 32. 

### 3.4. Methylation Analysis

Methylation analysis of the O-carbohydrate chain was carried out according to our previous study [37]. Briefly, after methylation (30 °C, 60 min), TFA hydrolysis (60 mg sodium borohydride 8 h, 100 °C), and acetic anhydride-pyridine acetylation (1 h, 100 °C), methylated alditol acetate derivatives of the O-carbohydrate chain were obtained and then analyzed by GC-MS (QP 2010, Shimadzu, Tokyo, Japan) using an RXI-5 SIL MS column (30 m × 0.25 mm × 0.25 µm, 120 °C initial temperature, 250 °C for 5 min, helium 1 mL/min).

### 3.5. NMR Analysis

NMR spectroscopy was performed to resolve the chemical structure of the O-carbohydrate chain [38]. Briefly, the O-carbohydrate chain of T-ES was dissolved in D_2_O and then freeze-dried, which was repeated three times. NMR was used to examine ^1^H NMR spectra, ^13^C NMR spectra, and DEPT135 ^1^D spectra and ^2^D spectra at 600 MHz.

### 3.6. HPLC-MS Spectrometry

The peptide chain of T-ES was analyzed according to the method in [39]. Briefly, T-ES was dissolved in Tris-HCl/dithiothreitol (100 mM, pH 8.5, 37 °C) for 2 h and subsequently in iodoacetamide for 15 min. After centrifugation, the precipitate was washed using Tris-HCl/NH_4_HCO_3_ and hydrolyzed using pancreatin. The peptide fragment was analyzed with an HPLC-MS system (Easy-nLC1200 and Q Exactive, Thermo, USA). The data were identified and quantified using PEAKS Studio 8.5 (Bioinformatics Solutions Inc., Waterloo, ON, Canada).

### 3.7. Animals’ Experiments

Male SAMP6 mice (12 w, 18 ± 2 g), one kind of senescence accelerated mice, were purchased from Beijing Zhishan Co. Ltd. (SCXK-2018-0010). The mice were housed in individual cages (12/12 h of light/dark, 23 ± 1 °C). The animal experiments were reviewed and approved by the Ethics Committee at Zhejiang Ocean University (No. 2021061). After 8 weeks of feeding, the SAMP6 mice were randomly divided into four groups (10 mice per group): model group (intragastric administration with 10 mL/kg of normal saline), ALN group (1 mg/kg of ALN), low dosage of T-ES group (T-ES-L, 40 mg/kg of T-ES), and high dosage of T-ES group (T-ES-H, 160 mg/kg of T-ES). Before the end of the experiments, the mice were fed in individual metabolism cages to collect urine. After 120 days of continuous feeding, the mice were anesthetized using diethyl ether and then sacrificed to obtain blood, femurs, and tibias for subsequent analysis.

### 3.8. Bone Mineral Density Measurement

Bone mineral density of the whole left-side femur and the whole left tibia were detected using a Dual-energy X-ray bone densitometer (GK99, I’acn, Italy). 

### 3.9. Bone Biomechanics Determination

Bone biomechanics of the left-side femur and tibia were determined using a universal material testing machine (5500, Instron, Boston, MA, USA). Briefly, bone samples were gently thawed from −80 °C to 25 °C, and the bone biomechanics were determined with a material testing machine (5 mm/min load velocity, 15 mm spacing) using the three-point bending method.

### 3.10. Bone Microstructure Observation

The distal end of the right-side femur and the proximal end of the right-side tibia in SAMP6 mice were fixed with neutral formalin, washed with running water, and subsequently scanned using a Micro-CT scanner (CT80, Scanco, Wangen-Brüttisellen, Switzerland), with 70 kV voltage, 114 µA electricity, and 10 µm distinguishability. The three-dimensional image was reestablished using the GPU NRecon Server (local) V1.7.4.2. Several parameters were assessed, including trabecular thickness, trabecular number, trabecular separation, connectivity density, structural model index, and cortical bone thickness.

### 3.11. Determination of Osteogenesis- and Bone Resorption-Related Markers in Serum and Urine

The blood was centrifuged to obtain the serum. Serum osteogenesis-related markers (BALP, PICP, and BGP) and bone resorption-related markers (RANKL, OPG, MMP-9, CTX) were determined using ELISA kits (Invitrogen, Carlsbad, CA, USA). Urinary Ca and P levels were measured using commercial kits (Jiancheng, Nanjing, China), and DPD content was determined using an ELISA kit (Invitrogen, Carlsbad, CA, USA), respectively.

### 3.12. Statistical Analysis

Data are expressed as the mean ± standard deviation (SD). One-way analysis of variance (ANOVA) followed by Dunnett’s T3 post-hoc test was performed for significant difference among the four groups with SPSS 21.0 software. Statistical significance was considered at *p* < 0.05.

## 4. Conclusions

In conclusion, a novel T-ES was extracted using KCl solution and purified using DEAE-Sepharose fast flow ion exchange chromatoe graphy followed by sephacryl S-300 gel filtration chromatography. The structure of the T-ES was as follows: the backbone carbohydrate →4)-β-D-GlcN-(1→3)-α-D-GalN-(1→3)-β-D-Glc-(1→2)-α-D-Gal-(1→2)-α-D-Gal-(1→3)-α-D-Man-(1→ with two branches of Neu5Ac(2→6)β-D-GlcN-(1→ and α-D-GalN-(1→ linking at o-4 in →2,4)-α-D-Gal-(1→; the peptide chain (Ala-Asp-Asn-Lys-Ser*-Met-Ile) connecting to the carbohydrate chain through O-glycosylation at the –OH of serine. Moreover, the T-ES exhibited significant anti-osteogenesis effects in SAMP mice.

## Figures and Tables

**Figure 1 marinedrugs-21-00573-f001:**
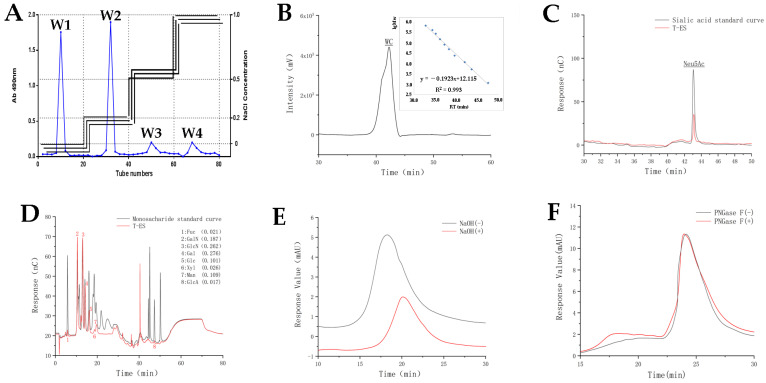
Isolation, purification, and general property analysis of T-ES. (**A**), Stepwise elution curve of T-ES on a DEAE-52 ion exchange column; (**B**), Purification of T-ES by Automatic Gel Purification System and its molecular weight; (**C**), HPLC analysis for Neu5Ac in T-ES; (**D**), GM analysis for monosaccharide in T-ES; (**E**), HPLC analysis for T-ES treated with/without NaOH; (**F**), HPLC analysis for T-ES treated with/without PNGase.

**Figure 2 marinedrugs-21-00573-f002:**
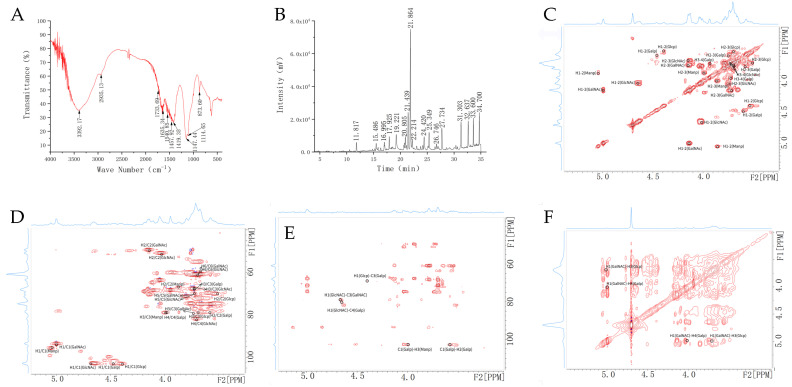
Structural characterization of carbohydrate chain in T-ES. (**A**), FT-IR analysis; (**B**), polysaccharide methylation; (**C**), HH-COSY spectrum; (**D**), HSQC spectrum; (**E**), HMBC spectrum; (**F**), NOESY spectrum.

**Figure 3 marinedrugs-21-00573-f003:**
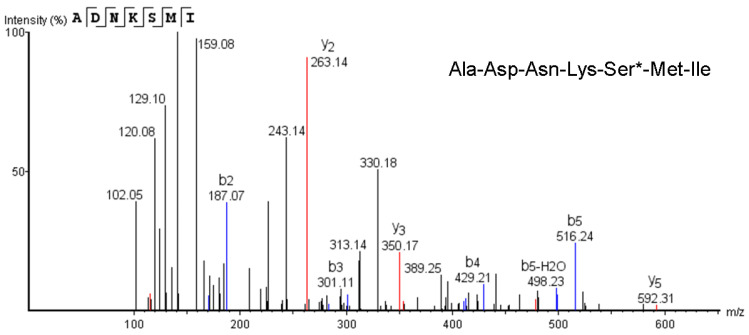
Two-stage mass spectrum of a peptide segment of T-ES. Ser* represents the O-glycosylation between Ser and sugars.

**Figure 4 marinedrugs-21-00573-f004:**
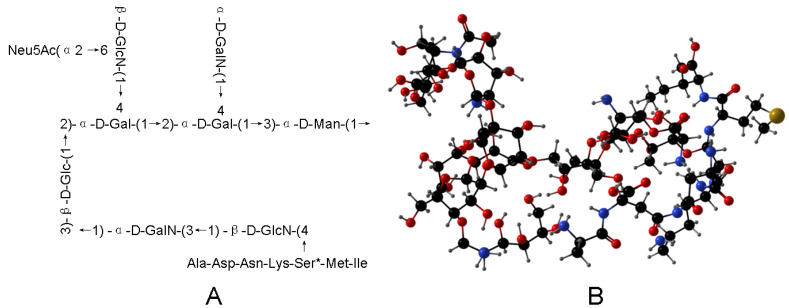
Chemical structure of T-ES. (**A**), T-ES structure; (**B**), the three-position structure of T-ES.

**Figure 5 marinedrugs-21-00573-f005:**
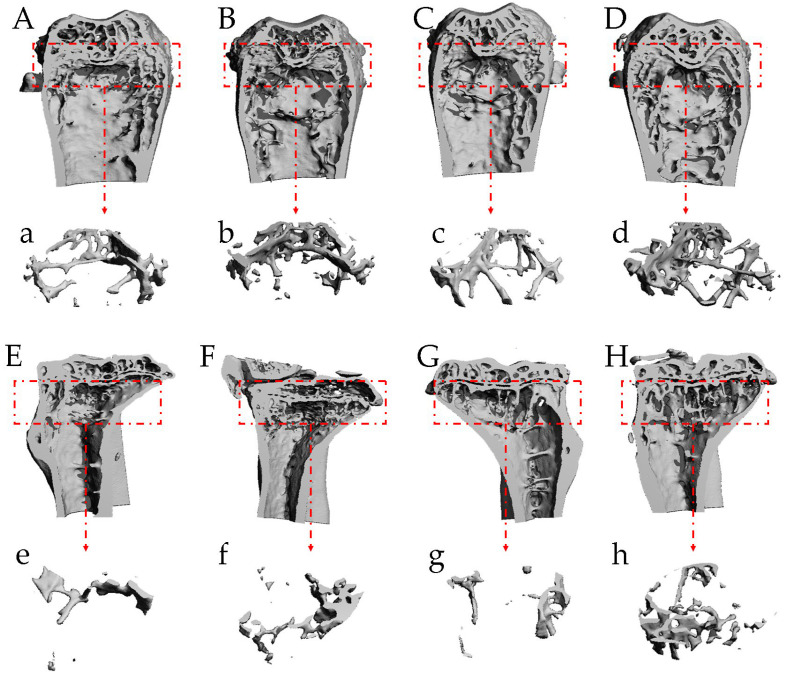
Effects of T-ES on the microstructure of femoral and tibial tissues. (**A**,**a**), femoral microstructure in the model group; (**B**,**b**), femoral microstructure in the ALN group; (**C**,**c**), femoral microstructure in the T-ES-L group; (**D**,**d**), femoral microstructure in the T-ES-H group; (**E**,**e**), tibial microstructure in the model group; (**F**,**f**), tibial microstructure in the ALN group; (**G**,**g**), tibial microstructure in the T-ES-L group; (**H**,**h**), tibial microstructure in the T-ES-H group.

**Table 1 marinedrugs-21-00573-t001:** Component characterization of monosaccharide in T-ES.

Sample	Monosaccharide Composition Proportion %								M_p_ (Da)	M_w_ (Da)	M_n_ (Da)
	Fuc	GalN	GlcN	Gal	Glc	Xyl	Man	GlcA			
T-ES	0.021	0.187	0.262	0.276	0.101	0.026	0.109	0.017	8481	9481	7221

**Table 2 marinedrugs-21-00573-t002:** Analysis of T-ES methylation.

RT	Methylated Sugar	Mass Fragments (*m*/*z*)	MOLAR RATIO	Type of Linkage
11.817	2,3,4-Me_3_-Fuc	43, 59, 72, 89, 101, 115, 117, 131, 175	0.027	Fuc-(1→
15.486	2,3-Me_2_-Xyl	43, 71, 87, 99, 101, 117, 129, 161, 189	0.024	→4)-Xyl-(1→
16.995	2,3,4,6-Me_4_-Glc	43, 71, 87, 101, 117, 129, 145, 161, 205	0.027	Glc-(1→
17.925	2,3,4,6-Me_4_-Man	43, 71, 87, 101, 117, 129, 145, 161, 205	0.030	Man-(1→
19.221	3,4,6-Me_3_-D-Glc-2-N	43, 87, 101, 117, 189, 205	0.054	GlcN-(1→
20.805	3,4,6-Me_3_-Man	43, 87, 129, 161, 189	0.013	→2)-Man-(1→
20.926	2,4,6-Me_3_-Glc	43, 87, 99, 101, 117, 129, 161, 173, 233	0.032	→3)-Glc-(1→
21.439	2,3,6-Me_3_-Glc	43, 87, 99, 101, 113, 117, 129, 131, 161, 173, 233	0.095	→4)-Glc-(1→
21.864	2,4,6-Me_3_-Gal	43, 87, 99, 101, 117, 129, 161, 173, 233	0.299	→3)-Gal-(1→
22.214	2,3,4-Me_3_-Glc	43, 87, 99, 101, 117, 129, 161, 189, 233	0.031	→6-Glc-(1→
24.42	2,6-Me_2_-Glc	43, 87, 97, 117, 159, 185	0.041	→3,4)-Glc-(1→
25.349	2,3-Me_2_-Man	43, 71, 85, 87, 99, 101, 117, 127, 159, 161, 201	0.045	→4,6)-Man-(1→
26.746	2,3-Me_2_-Glc	43, 71, 85, 87, 99, 101, 117, 127, 159, 161, 201	0.017	→4,6)-Glc-(1→
27.013	2,4-Me_2_-Glc	43, 87, 117, 129, 159, 189, 233	0.014	→3,6)-Glc-(1→
27.734	2,3-Me_2_-Gal	43, 71, 85, 87, 99, 101, 117, 127, 159, 161, 201, 261	0.058	→4,6)-Gal-(1→
31.303	4,6-Me_2_-D-Gal-2-N	45, 73, 87, 99, 117, 129, 173, 189, 233	0.060	→3)-GalN-(1→
32.637	4,6-Me_2_-D-Glc-2-N	45, 73, 87, 99, 117, 129, 173, 189, 233	0.053	→3)-GlcN-(1→
33.600	4-Me_1_-D-Gal-2-N	43, 74, 116, 129, 143, 158, 233	0.039	→3,6)-GalN-(1→
34.700	4-Me_1_-D-Glc-2-N	43, 74, 116, 129, 143, 158, 233	0.040	→3,6)-GlcN-(1→

**Table 3 marinedrugs-21-00573-t003:** Effects of T-ES on osteoporosis-related parameters in SAMP6 mice.

Tissues	Parameters	Model	ALN	T-ES-L	T-ES-H
Femur	Bone density (mg/cm^2^)	82.35 ± 9.03 ^a^	126.54 ± 13.28 ^b^	90.48 ± 10.07 ^a^	115.91 ± 11.88 ^b^
	Max. load (N)	12.31 ± 0.13 ^a^	24.52 ± 0.30 ^b^	15.58 ± 0.22 ^c^	22.77 ± 0.29 ^b^
	Max. deflection (mm)	0.56 ± 0.04 ^a^	0.92 ± 0.10 ^b^	0.63 ± 0.0.07 ^a^	1.02 ± 0.11 ^b^
	Trabecular thickness (µm)	41.25 ± 3.68 ^a^	96.42 ± 8.57 ^b^	60.47 ± 5.40 ^c^	100.54 ± 9.63 ^b^
	Trabecular number (1/mm)	2.29 ± 0.05 ^a^	2.91 ± 0.03 ^b^	2.45 ± 0.07 ^a^	3.56 ± 0.08 ^c^
	Trabecular separation (mm)	0.42 ± 0.03 ^a^	0.29 ± 0.02 ^b^	0.38 ± 0.02 ^a^	0.26 ± 0.02 ^b^
	Conectivity density (1/mm^3^)	25.44 ± 1.58 ^a^	37.87 ± 1.85 ^b^	27.06 ± 2.07 ^a^	39.29 ± 2.14 ^b^
	Structureal model index	1.87 ± 0.19 ^a^	0.95 ± 0.09 ^b^	1.74 ± 0.20 ^a^	0.81 ± 0.08 ^b^
	Cortical bone thickness (mm)	0.15 ± 0.01 ^a^	0.19 ± 0.01 ^b^	0.16 ± 0.00 ^a^	0.19 ± 0.01 ^b^
Tibia	Bone density (mg/cm^2^)	51.26 ± 4.89 ^a^	114.07 ± 12.36 ^b^	79.40 ± 8.55 ^c^	109.92 ± 15.48 ^b^
	Max. load (N)	10.37 ± 0.10 ^a^	18.64 ± 0.23 ^bc^	17.71 ± 0.19 ^b^	20.45 ± 0.23 ^cd^
	Max. deflection (mm)	0.69 ± 0.08 ^a^	1.32 ± 0.14 ^b^	1.04 ± 0.09 ^c^	1.25 ± 0.11 ^b^
	Trabecular thickness (µm)	58.39 ± 5.26 ^a^	83.24 ± 9.10 ^b^	72.61 ± 9.02 ^ab^	85.90 ± 7.47 ^b^
	Trabecular number (1/mm)	2.11 ± 0.26 ^a^	3.25 ± 0.27 ^b^	2.76 ± 0.19 ^ab^	3.48 ± 0.22 ^b^
	Trabecular separation (mm)	0.60 ± 0.04 ^a^	0.36 ± 0.04 ^b^	0.45 ± 0.03 ^c^	0.31 ± 0.05 ^bc^
	Conectivity density (1/mm^3^)	9.14 ± 0.88 ^a^	21.52 ± 4.31 ^b^	12.39 ± 2.24 ^a^	23.43 ± 3.16 ^b^
	Structureal model index	3.78 ± 0.42 ^a^	1.86 ± 0.20 ^bc^	2.22 ± 0.18 ^b^	1.55 ± 0.17 ^c^
	Cortical bone thickness (mm)	0.14 ± 0.01 ^a^	0.23 ± 0.01 ^b^	0.19 ± 0.01 ^c^	0.24 ± 0.01 ^b^
Serum	BALP (ng/mL)	6.39 ± 0.74 ^a^	8.24 ± 0.78 ^b^	6.81 ± 0.70 ^a^	8.83 ± 0.90 ^b^
	PICP (ng/mL)	10.68 ± 1.00 ^a^	15.58 ± 1.34 ^b^	14.22 ± 1.07 ^b^	16.21 ± 1.37 ^b^
	BGP (ng/mL)	12.03 ± 1.44 ^a^	15.62 ± 1.37 ^b^	16.98 ± 1.45 ^bc^	17.87 ± 1.69 ^c^
	RANKL (pg/mL)	18.04 ± 1.55 ^a^	24.87 ± 2.61 ^b^	21.51 ± 1.88 ^c^	25.14 ± 2.10 ^b^
	OPG (pg/mL)	13.36 ± 1.07 ^a^	7.94 ± 0.81 ^b^	12.68 ± 1.45 ^a^	7.57 ± 0.67 ^b^
	OPG/RANKL	0.73 ± 0.07 ^a^	0.32 ± 0.04 ^b^	0.63 ± 0.09 ^a^	0.31 ± 0.05 ^b^
	Cath-k (ng/mL)	4.04 ± 0.31 ^a^	2.83 ± 0.24 ^b^	3.67 ± 0.24 ^a^	2.71 ± 0.28 ^b^
	MMP-9 (ng/mL)	42.52 ± 3.16 ^a^	32.44 ± 2.14 ^b^	39.51 ± 2.30 ^c^	31.50 ± 2.59 ^b^
	CTX-I (ng/mL)	18.90 ± 1.73 ^a^	15.23 ± 1.23 ^bc^	16.04 ± 0.95 ^b^	14.34 ± 1.11 ^c^
Urine	DPD (mmol/L)	0.23 ± 0.01 ^a^	0.14 ± 0.01 ^b^	0.15 ± 0.01 ^b^	0.11 ± 0.01 ^b^
	Ca (mmol/L)	0.33 ± 0.06 ^a^	0.14 ± 0.01 ^b^	0.29 ± 0.03 ^a^	0.16 ± 0.02 ^b^
	P (mmol/L)	4.77 ± 0.30 ^a^	2.04 ± 0.18 ^b^	3.45 ± 0.33 ^c^	2.26 ± 0.25 ^b^

Note: Data are presented as mean ± S.D. (*n* = 10). Multiple comparisons were performed using one way ANOVA. Different lowercases represented significant difference (*p* < 0.05) between groups. ALN, alendronate; T-ES, sialoglycopeptide from tuna eggs; BALP, bone alkaline phosphatase; PICP, procollagen type I C-terminal peptide; BGP, bone gla protein; RANKL, receptor activator of nuclear factor-κ B ligand; OPG, osteoprotegerin; MMP-9, matrix metallo protein 9; CTX-I, C-terminal telopeptide of type I collagen; DPD, deoxypyridinoline; Ca, calcium; P, phosphorus.

## Data Availability

The data is unavailable due to privacy.

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
