# Peer review of "Structural Characterization and Anti-Osteoporosis Effects of a Novel Sialoglycopeptide from Tuna Eggs"

_marinedrugs, 2023, doi:10.3390/md21110573_

Round 1
Reviewer 1 Report
Comments and Suggestions for Authors
I have reviewed the manuscript titled “Structural characterization and anti-osteoporosis effects of a novel sialoglycopeptide from tuna eggs”. In its current form, I cannot recommend the manuscript for publication. The description of the methodology is highly inadequate. Since this work describes a new product with possible therapeutic implications, all extraction, purification, and analytical procedures must be clearly described to be replicable.
Many references are missing (empty []), while others seem to be chosen completely at random: in the introduction, ref. 1,2,4,6; in the results, ref. 24, and so on.
Results:
L182-183. "Considering that DXA measures BMD (bone mineral density), stating 'Bone mineral density is an important indicator to assess bone mineral density' is redundant."
L200 This sentence seems to be redundant.
Table 3 Cortical bone thickness of 0.14-0.23 μm ??
Methods:
Section: 3.1 specify the temperature, time, and other conditions (e.g., pH) during the various processes, like lyophilization, extraction, hydrolysis, and centrifugation, for better reproducibility. It's unclear what "differential refraction detector" means in this context.
Section 3.2 Again, for better reproducibility some additional details should be provided. For example, for the HPLC and ion chromatograph methods, it's essential to provide details like mobile phase, column temperature, flow rate, and detection wavelength. For analysis of glycosylation type, it would be helpful to provide the concentration of the PNGase F enzyme and NaOH used, along with the temperature and time of hydrolysis.
Section 2.3 It would be helpful to specify the conditions (e.g., resolution, number of scans) for the FT-IR analysis. The term "troche" is uncommon in this context.
Section 3.4 "Briefly, after methylation, TFA hydrolysis..." specify the reagents and conditions (e.g., time, temperature) for the methylation, hydrolysis, and acetylation for clarity and reproducibility.
Section 3.5 Mentioning the solvent used for the NMR spectra might be beneficial. It's also a standard to specify the magnetic field strength (e.g., 600 MHz) in relation to either proton or carbon NMR.
Section 3.6 conditions used during the dissolution in Tris-HCl/dithiothreitol (e.g., temperature, concentration).
Section 3.7 "After feeding 8 w" – Please clarify whether this is "after 8 weeks of feeding."
Section 3.8 DXA measures “bone mineral density”, not “bone density” (bulk). How this measurement was performed? For whole bone, or just a section ?
Section 3.9 This section, in its current form, is entirely lacking in detail. There is no information provided about the type of test performed (3-point bending? 4-point bending? Compression? Other?), the parameters measured, or the conditions under which the mechanical tests were conducted (e.g., load rate, etc.), preparation of the bones (fresh? Frozen?) .
Section 3.10: This section faces the same problem as mentioned earlier. As it stands, it is severely lacking in specifics. What parameters were assessed? How exactly were the bones prepared (simply fixing in formalin is not comprehensive)? Which portions of the bones were scanned? What were the scanning conditions, including voxel size? Additionally, which software was utilized for analysis? “suffered” ? did you mean “fixed” ? The term "Micro-CT scanister" is not standard. It should be "Micro-CT scanner."
Section 3.11: Is it possible to measure DPD using "commercial kits" instead of an ELISA kit? I'm inclined to believe that the CTX measured in this study refers to the "C-terminal telopeptide of type 1 collagen," a marker for bone resorption, rather than "Cyclophosphamide," a cytotoxic agent typically utilized in cancer treatments.
Section 3.12. After performing ANOVA, a Student’s t-test was applied. This is not standard. If you are comparing more than two groups, a post-hoc test should be used.
Author Response
Wednesday, 18 October 2023
Marine Drugs
Dear reviewer,
Thank you for your effort on our manuscript, namely “Structural characterization and anti-osteoporosis effects of a novelsialoglycopeptide from tuna eggs”. We also appreciate the meticulous review provided by your expert. Our responses to the comments are as follows.
Reviewer 1
I have reviewed the manuscript titled “Structural characterization and anti-osteoporosis effects of a novel sialoglycopeptide from tuna eggs”. In its current form, I cannot recommend the manuscript for publication. The description of the methodology is highly inadequate. Since this work describes a new product with possible therapeutic implications, all extraction, purification, and analytical procedures must be clearly described to be replicable.
Many references are missing (empty []), while others seem to be chosen completely at random: in the introduction, ref. 1,2,4,6; in the results, ref. 24, and so on.
Answer: Thank you very much for the suggestions. We have added the missing references.
Results:
L182-183. "Considering that DXA measures BMD (bone mineral density), stating 'Bone mineral density is an important indicator to assess bone mineral density' is redundant."
Answer: Thank you very much for the suggestions. We have revised the sentence as “Bone mineral density is the golden standard to osteoporosis of early diagnosis, therapeutic, and prognosis”.
L200 This sentence seems to be redundant.
Answer: Thank you very much for the suggestions. We think the sentence is important because it accounts for the significance of bone biomechanical properties.
Table 3 Cortical bone thickness of 0.14-0.23 μm ??
Answer: Thank you very much for the suggestions. We have corrected the units of cortical bone thickness as “mm”.
Methods:
Section: 3.1 specify the temperature, time, and other conditions (e.g., pH) during the various processes, like lyophilization, extraction, hydrolysis, and centrifugation, for better reproducibility. It's unclear what "differential refraction detector" means in this context.
Answer: Thank you very much for the suggestions. We have added the details of various processes.
Section 3.2 Again, for better reproducibility some additional details should be provided. For example, for the HPLC and ion chromatograph methods, it's essential to provide details like mobile phase, column temperature, flow rate, and detection wavelength. For analysis of glycosylation type, it would be helpful to provide the concentration of the PNGase F enzyme and NaOH used, along with the temperature and time of hydrolysis.
Answer: Thank you very much for the suggestions. We have added the details as your suggestions.
Section 2.3 It would be helpful to specify the conditions (e.g., resolution, number of scans) for the FT-IR analysis. The term "troche" is uncommon in this context.
Answer: Thank you very much for the suggestions. We have specified the conditions as your suggestions.
Section 3.4 "Briefly, after methylation, TFA hydrolysis..." specify the reagents and conditions (e.g., time, temperature) for the methylation, hydrolysis, and acetylation for clarity and reproducibility.
Answer: Thank you very much for the suggestions. We have added the details as your suggestions.
Section 3.5 Mentioning the solvent used for the NMR spectra might be beneficial. It's also a standard to specify the magnetic field strength (e.g., 600 MHz) in relation to either proton or carbon NMR.
Answer: Thank you very much for the suggestions. The solvent and the magnetic field strength had been mentioned in methods.
Section 3.6 conditions used during the dissolution in Tris-HCl/dithiothreitol (e.g., temperature, concentration).
Answer: Thank you very much for the suggestions. We have revised it as your advices.
Section 3.7 "After feeding 8 w" – Please clarify whether this is "after 8 weeks of feeding."
Answer: Thank you very much for the suggestions. We have changed “after feeding 8 w” to “after 8 weeks of feeding”.
Section 3.8 DXA measures “bone mineral density”, not “bone density” (bulk). How this measurement was performed? For whole bone, or just a section ?
Answer: Thank you very much for the suggestions. We have changed “bone density” to “bone mineral density”.
Section 3.9 This section, in its current form, is entirely lacking in detail. There is no information provided about the type of test performed (3-point bending? 4-point bending? Compression? Other?), the parameters measured, or the conditions under which the mechanical tests were conducted (e.g., load rate, etc.), preparation of the bones (fresh? Frozen?) .
Answer: Thank you very much for the suggestions. We have revised it as your advices.
Section 3.10: This section faces the same problem as mentioned earlier. As it stands, it is severely lacking in specifics. What parameters were assessed? How exactly were the bones prepared (simply fixing in formalin is not comprehensive)? Which portions of the bones were scanned? What were the scanning conditions, including voxel size? Additionally, which software was utilized for analysis? “suffered” ? did you mean “fixed” ? The term "Micro-CT scanister" is not standard. It should be "Micro-CT scanner."
Answer: Thank you very much for the suggestions. We have revised it as your suggestions.
Section 3.11: Is it possible to measure DPD using "commercial kits" instead of an ELISA kit? I'm inclined to believe that the CTX measured in this study refers to the "C-terminal telopeptide of type 1 collagen," a marker for bone resorption, rather than "Cyclophosphamide," a cytotoxic agent typically utilized in cancer treatments.
Answer: Thank you very much for the suggestions. We have revised it as your suggestions.
Section 3.12. After performing ANOVA, a Student’s t-test was applied. This is not standard. If you are comparing more than two groups, a post-hoc test should be used.
Answer: Thank you very much for the suggestions. We have reanalyzed using a post-hoc test.
Thank you for your consideration. I look forward to hearing from you.
Best wishes
Yours sincerely,
Shiwei Hu
Reviewer 2 Report
Comments and Suggestions for Authors

Comments on the Quality of English Language
Author Response
Wednesday, 18 October 2023
Marine Drugs
Dear reviewer,
Thank you for your effort on our manuscript, namely “Structural characterization and anti-osteoporosis effects of a novelsialoglycopeptide from tuna eggs”. We also appreciate the meticulous review provided by your expert. Our responses to the comments are as follows.
Reviewer 2
The manuscript “Structural characterization and anti-osteoporosis effects of a novel sialoglycopeptide from tuna eggs” describes preparation of sialoglycopeptide from tuna eggs and applying the peptide in senescence mice model to evaluate the anti-osteoporosis effects. The authors characterized the prepared sialoglycopeptide from a wide range of analytical procedures such as general properties by HPLC, FT-IR Spectroscopy Analysis, Methylation analysis by GC-MS, NMR analysis, HPLC-MS spectrometry etc. The effects of administered sialoglycopeptide in SAMP mice was evaluated and determined by bone density measurement, bone biomechanics and microstructure measurement, osteogenesis and bone resorption parameters in serum and urine etc. The authors found a significant antiosteogenesis effect in mice. The theme of the study is impactful for developing antiosteogenesis drugs and for the readers of the field. Additionally, the research is well designed and writing is also sound. However, there are some errors and limitation, (e.g., discussions)
that are needed to improve before the manuscript accepted in Marine Drugs.
Specific comments:
- Line 10; The authors should specify the name of fish from which the eggs was collected.
Answer: Thank you very much for the suggestions. The fishes include several species, including rainbow trout, medaka fish, pacific herring, and Gadus morhua. therefore, we changed “specify” to “several”.
Line 21; Please write elaborately SAMP in first use.
Answer: Thank you very much for the suggestions. We have described SAMP as “senescence-accelerated mouse prone (SAMP6)”.
- Line 22; I suggest to write “the present study result” instead of “our result”.
Answer: Thank you very much for the suggestions. We have revised “our results” to “the present study results”.
- Line 32; Could you please add some reasons for osteoporosis problem.
Answer: Thank you very much for the suggestions. The purpose of our study is to investigate the effects of T-ES on osteoporosis, rather than the cause of osteoporosis. Therefore, we believe that the detriment to human of osteoporosis is described.
- Line 54; “Our previous experiments proved that the concentration of sialoglycopeptide from tuna eggs (T-ES) accounts for 4%” please add reference for
the statement.
Answer: Thank you very much for the suggestions. The data was only detected, but not published.
- Between line 56 and 57; the authors are highly emphasized to state few sentences
regarding the necessity of conducting present study.
Answer: Thank you very much for the suggestions. We have added a sentence to explain the necessity of the present study as “This research will provide theoretical basis for application in prevention and treatment of senile opteoporosis”.
- Figure 1 & 2; The figures are not clearly visible. Please enlarge the figure sixe and
resolution for the easy understanding of the readers.
Answer: Thank you very much for the suggestions. We have revised the figures.
- 2.6. T-ES improved bone microstructure; The mechanisms should be discussed in this section. Additionally, the authors should improve discussion by comparing with the previous study of related research.
Answer: Thank you very much for the suggestions. We have added the discussion as your suggestions.
- The biochemical composition of fish eggs are related to their developmental stages.
Therefore, it is important to me stating the developmental stage of fish eggs used in
this study.
Answer: Thank you very much for the suggestions. We have revised “tuna eggs” to “mature tuna eggs”.
- Line 252; The authors should specify the species of tuna used for this study.
Answer: Thank you very much for the suggestions. We have revised “tuna eggs” to “striped tuna eggs”.
- Line 263; The process for removal of lipids should be described more detail.
Answer: Thank you very much for the suggestions. We have revised the sentence as “Homogenized fish eggs were removed lipids with ethyl alcohol (1:10, m/V, 25℃, 8 h)”.
- Line 255; The hydrolysis process of sialoglycopeprotein is a key step to obtain sialoglycopeptide successfully. So, I recommend describing the method to facilitate reproducibility to the readers or add suitable reference of previous report.
Answer: Thank you very much for the suggestions. We have added the suitable reference [14] to support our method.
- Line 291: The reference is missing. Please add this.
Answer: Thank you very much for the suggestions. We have added the reference.
Thank you for your consideration. I look forward to hearing from you.
Best wishes
Yours sincerely,
Shiwei Hu
Round 2
Reviewer 1 Report
Comments and Suggestions for Authors
I must maintain my negative review of this work.
The authors claimed to have corrected and provided the appropriate references (e.g., 1,2,4,6), but they remain unchanged.
Moreover, I requested that the authors provide more detailed information regarding the preparation and realization of various technological processes. The citation provided in line 256, reference [14], which is purported to describe the details of the hydrolysis process, does not explain the course of this process and refers to another work.
In Section 3.3, contrary to the authors' assurances, the word "troche" was not corrected to "pellet". Information on the resolution of the FTIR spectrum and the number of scans conducted is still missing.
In Section 3.8, no changes were made to the manuscript. How was this measurement performed? Was it for the whole bone or just a section?
In Section 3.9, the provided description does not answer some of my queries: there is no information provided about the type of test performed (3-point bending? 4-point bending? Compression? Shear? Other?).
Section 3.10. As previously mentioned, the provided description doesn't answer some of my queries: which software was utilized for analysis? What parameters were assessed? I also believe that the distal femur and proximal tibia were scanned, not the distal parts of both bones.
Section 3.12: So, which post-hoc test was used?
In line 178, the statement "Bone biomechanical properties are of great significance to evaluate the repair and therapeutic of osteoporosis" is considered redundant as it is imprecise. Bone biomechanical properties determine its functional properties, not the evaluation of repair – the strength tests of the bone serve this purpose, not its functional features. Therefore, this debatable statement should either be removed or corrected.
In line 252, can the authors specify from which fishing grounds (areas) the tuna eggs were obtained?
In lines 329, 330, and 331, please correct "parameters" to "markers".
In line 333, was DPD measured using commercial kits or an ELISA kit? (please provide the correct answer).
In line 360, I believe OPG stands for "osteoprotegerin", not "Osteoclastogenesis inhibitory factor" (commonly abbreviated as OCIF).
Author Response
Monday, 23 October 2023
Marine Drugs
Dear reviewer,
Thank you for your effort on our manuscript, namely “Structural characterization and anti-osteoporosis effects of a novelsialoglycopeptide from tuna eggs”. We also appreciate the meticulous review provided by your expert. Our responses to the comments are as follows.
Reviewer 1
I must maintain my negative review of this work.
The authors claimed to have corrected and provided the appropriate references (e.g., 1,2,4,6), but they remain unchanged.
Answer: Thank you very much for the suggestions. We have changed the references 1, 2, 4, and 6, and have revised the sentences according to the references.
Moreover, I requested that the authors provide more detailed information regarding the preparation and realization of various technological processes. The citation provided in line 256, reference [14], which is purported to describe the details of the hydrolysis process, does not explain the course of this process and refers to another work.
Answer: Thank you very much for the suggestions. We have described the details of the hydrolysis process as “The sialoglycoprotein was hydrolyzed using 1.5% alkaline protease in a water bath oscillator at 50℃ for 2 h, and then terminated enzymolysis at 100℃ for 10 min. After centrifugation at 7500×g for 15 min, the liquid supernatant was concentrated and lyophilized to collect crude T-ES”.
In Section 3.3, contrary to the authors' assurances, the word "troche" was not corrected to "pellet". Information on the resolution of the FTIR spectrum and the number of scans conducted is still missing.
Answer: Thank you very much for the suggestions. We have revised "troche" to "pellet". And we have added the resolution of the FTIR spectrum of 4 cm-1 and the number of scans 32.
In Section 3.8, no changes were made to the manuscript. How was this measurement performed? Was it for the whole bone or just a section?
Answer: Thank you very much for the suggestions. We have revised the bone as “the whole left side femur and the whole left tibia”.
In Section 3.9, the provided description does not answer some of my queries: there is no information provided about the type of test performed (3-point bending? 4-point bending? Compression? Shear? Other?).
Answer: Thank you very much for the suggestions. We have added the information as “using the three-point bending method”.
Section 3.10. As previously mentioned, the provided description doesn't answer some of my queries: which software was utilized for analysis? What parameters were assessed? I also believe that the distal femur and proximal tibia were scanned, not the distal parts of both bones.
Answer: Thank you very much for the suggestions. We have corrected the distal parts of both bones as “the distal end of right side femur and the proximal end of right side tibia”. We also added the details as “The three-dimensional image was reestablished using GPU NRecon Server (local) V1.7.4.2. Several parameters were assessed, including trabecular thickness, trabecular number, trabecular separation, concetivity density, structureal model index, and cortical bone thickness.”
Section 3.12: So, which post-hoc test was used?
Answer: Thank you very much for the suggestions. We have revised it as “Dunnett’s T3 post-hoc test”.
In line 178, the statement "Bone biomechanical properties are of great significance to evaluate the repair and therapeutic of osteoporosis" is considered redundant as it is imprecise. Bone biomechanical properties determine its functional properties, not the evaluation of repair – the strength tests of the bone serve this purpose, not its functional features. Therefore, this debatable statement should either be removed or corrected.
Answer: Thank you very much for the suggestions. We have revised the sentence as “Bone biomechanical properties are of great significance to evaluate osteoporosis”.
In line 252, can the authors specify from which fishing grounds (areas) the tuna eggs were obtained?
Answer: Thank you very much for the suggestions. We have added the details as “which were captured from South Pacific Ocean”.
In lines 329, 330, and 331, please correct "parameters" to "markers".
Answer: Thank you very much for the suggestions. We have corrected "parameters" to "markers".
In line 333, was DPD measured using commercial kits or an ELISA kit? (please provide the correct answer).
Answer: Thank you very much for the suggestions. We have revised the sentences as “Urinary Ca and P levels were measured using commercial kits (Jiancheng, Nanjing, China), and DPD content was determined using ELISA kit (Invitrogen, Carlsbad, CA, USA), respectively”.
In line 360, I believe OPG stands for "osteoprotegerin", not "Osteoclastogenesis inhibitory factor" (commonly abbreviated as OCIF).
Answer: Thank you very much for the suggestions. We have corrected the full name of OPG as osteoprotegerin.
Thank you for your consideration. I look forward to hearing from you.
Best wishes
Yours sincerely,
Shiwei Hu
Reviewer 2 Report
Comments and Suggestions for Authors
Thank you very much for the improvement.
Comments on the Quality of English Language
Minor grammatical and language editing are recommended.
Author Response
Thank you very much for your help to our manuscript. And we have revised the grammatical and language.
Round 3
Reviewer 1 Report
Comments and Suggestions for Authors
All major issues highlighted in the initial review have been adressed.